# Meaning without reference in large language models

## Abstract

The widespread success of large language models (LLMs) has been met with skepticism that they possess anything like human concepts or meanings. Contrary to claims that LLMs possess no meaning whatsoever, we argue that they likely capture important aspects of meaning, and moreover work in a way that approximates a compelling account of human cognition in which meaning arises from *conceptual role*. Because conceptual role is defined by the relationships between internal representational states, meaning cannot be determined from a model's architecture, training data, or objective function, but only by examination of how its internal states relate to each other. This approach may clarify why and how LLMs are so successful and suggest how they can be made more human-like.

LARGE language models (LLMs) have begun to display an array of competencies that were long thought to be out of the reach of neural networks. At the same time, critics have been vocal that existing methods, model structures, and training paradigms will be insufficient for anything like human language use. In particular, it is argued that LLMs will never achieve "meaning" or "understanding" due to either the objective function they optimize, the format of their internal representations, or the type of training data that they receive. This short comment aims to contest the claims that LLMs are necessarily incapable of acquiring meaning, and suggest that LLM meanings may already be similar to human-like meaning in many (but not all) ways. We argue that LLMs have likely already achieved some key aspects of meaning which, while imperfect, mirror the way meanings work in cognitive theories, as well as approaches to the philosophy of language.

LLMs are trained to predict words from massive datasets of text from the internet. They typically contain billions of parameters that are jointly optimized—at great computational, energy, and financial expense [Bender et al., 2021]—to make predictions about word occurrence from surrounding context. This setup differs from human language acquisition in data scale and format. On the other hand, the core objective of word prediction is a central piece of human language processing [e.g. Altmann and Kamide, 1999, Hale, 2001, Levy, 2008] and has long been shown capable of providing a learning signal from which linguistic structures and semantic categories can emerge [Elman, 1990].

A key debate in recent literature has been about *what* models trained in this manner come to know. A prominent view is that models trained only on text cannot acquire realistic meanings because they lack reference, or connection to objects in the real world. Bender and Koller [2020] illustrate this with an "octopus test." They imagine an octopus that learns to use words correctly by eavesdropping on a conversation between two people on land. If the octopus has no access to the referents of the words then there are gaps in its meaning for the words. For example, if the octopus must suddenly determine which *object* is a coconut, then its expertise in using the word "coconut" won't help. Its knowledge of co-occurrence statistics between "coconut" and other words won't help either since it also knows nothing about the referents of other words. The octopus simply does not have the required meanings to find the coconut. Humans learners don't have this problem because their input—and consequently representations—are tied to real-world referents. Bender & Koller's position is that no

amount of predictive linguistic savvy can give models that are trained on text alone the knowledge of reference they need to acquire meanings.

**Meaning and reference**   The octopus test assumes that reference determines meaning, but in fact cognitive scientists and philosophers have found a variety of problems with this view. One is that there are many terms that are meaningful to us but have no discernible referent at all, such as abstract words like "justice" and "wit." People can think of new concepts like "aphid-sized accordion" that don't exist (thus no referent), or even terms that have no *possible* referent like "perpetual motion machine" or "imaginary cup of tea." We can think of concepts like "king of San Francisco" that pick out nobody, but are at least meaningful enough to reason about (for example: "If there was a King of San Francisco, he'd live in The Presidio."). Other examples show reference can be quite decoupled from meaning. Terms like "treaty" and "contract" are often thought to have a concrete referent, but what is important is actually an abstract entity: a treaty is still valid if the piece of paper is destroyed. Frege's example is of the "morning star" and the "evening star." Both are terms for the planet Venus, but were once conceived of as different entities without knowing that they are the same object.

This problem is not solely an issue with abstract concepts. Many concrete terms seem not to get their most important semantic properties from reference either. Consider an example like a "postage stamp." Everyone can conjure up an image of a typical, physical, postage stamp. But with further scrutiny, none of the concrete features of typical stamps seem necessary. We can imagine a country whose postage stamps were made of clear glass, for example, or stamps that were microscopic so that they could not be seen, or stamps that were paid for and tracked entirely online, or others that were larger than a house (for mailing very large packages). We could think of postage stamps that were *RFID* tags that went inside the envelope. Maybe intelligent ants would use postage stamps that were pheromones rather than paper; in the future we might have postage stamps that sprout wings and fly your letter to its intended location. Already we have stamps that only have barcodes and stamps that you can draw yourself. You know the term "postage stamp" but there is no way that you could have considered all of the possible referents yet, so reference cannot be what determines the concept.

Similar arguments were made by Wittgenstein [1953] when considering the concept of "water", leading him to conclude that reference plays little or no role in determining meaning in the general case. One way to explain these observations is to assume that our meaning for terms like "postage stamp" (or "water") may be primarily determined by the role these concepts play in some greater mental theory. Very roughly, people call something a "postage stamp" if you pay for it and then attach it to a letter in order to have the letter to be delivered. In this view, the meaning of the word is intrinsically intertwined with other concepts like "payment", "letter" and "delivery." The interrelation is key [see, e.g. Deacon, 1998, Santoro et al., 2021] because when the associated terms shift in meaning even slightly (e.g. credit cards were developed as wholly new a form of "payment") the meaning of "postage stamp" comes along for the ride (we know right away that they can be paid for by a credit card). Such relationships between concepts are *the* essential, defining, aspects of meaning and, in fact, possessing the appropriate relationships allows you to determine the reference. This view makes sense of the puzzling examples above.

**Conceptual role theory**   In philosophy of mind, versions of this approach go by the name "conceptual role theory." Following Block [1998], consider statements in physics like $f = m \cdot a$. This equation is not exactly a definition of $f$ (*force*), nor is it a definition of *mass* ($m$) or *acceleration* ($a$), but, is a statement about the interrelationship of $f$, $m$, and $a$. Most of us can't give much more detail about the ultimate physical reference of these terms—forces have something to do with interacting elementary particles (whatever those are), masses have something to do with the Higgs field (whatever that is)—but our thoughts about $f = m \cdot a$ certainly do not seem meaningless. Many believe that conceptual roles are one of the most promising ways to characterize human concepts *in general* [for an overview of this and competing theories, see, e.g., Margolis and Laurence, 1999]. Murphy and Medin [1985] for example argue that our organization of categories is based on entire theories of structured conceptual domains (rather than simple features or similarities), an idea they trace to Quine [1977]. Murphy and Medin note how we might reflexively consider a composite category such as "prime numbers or apples" to be an unnatural or incoherent set of entities. However, if we know someone called Wilma who is a number theorist grew up on an apple farm, then the category "topics of conversation with Wilma", involving the same constituent entities, seems perfectly reasonable. What is a natural category or concept depends on our mental conception of how the underlying pieces relate, and concepts can even be assembled fluidly, in an ad hoc manner or context-dependent manner

[Barsalou, 1983, Casasanto and Lupyan, 2015]. Theories in cognition have also been explored in learning models [e.g. Goodman et al., 2011, Ullman et al., 2012] and experimentally shown to shape how children explore the world [e.g. Gopnik et al., 1999, Gopnik and Schulz, 2004, Bonawitz et al., 2012].

**Conceptual role in LLMs**  If anything like this view is correct, then the search for meaning in learning models—or brains—should focus on understanding the way that the systems' internal representational states *relate to each other*. Once a learning model finds it probable that "postage stamps" are "affixed" to "letters" so they can be "delivered," then it has acquired some important pieces of conceptual role for these terms. It would not be possible to conclude anything about what meanings a system does and does not possess from its training data or architecture because these may not be informative about how the internal states relate to each other.

Relations between internal states have been long emphasized in cognitive theories [Shepard and Chipman, 1970, Deacon, 1998, Fodor and Pylyshyn, 1988], for example early attempts to discover the geometry of psychological space [Shepard, 1980] and more recent analyses of brain data based on representational similarity [Kriegeskorte et al., 2008]. Elman [2004] argues for a closely related view of the mental lexicon in which a word's meaning is the effect it has on other mental states. In deep learning models, the relational geometry of vector representations have been examined for instance in analogy problems [Mikolov et al., 2013], match to human similarities [Hill et al., 2017], and encoding of humanlike gradient distinctions [Vulić et al., 2017]. Grand et al. [2022] show that semantic embeddings from these models capture gradient scales of multiple features, like from "small" to "big" or "safe" to "dangerous." It is even possible to align the word representations acquired by text-based models across languages to translate between them effectively with no prior knowledge of which words or phrases should have the same meaning [Lample et al., 2017]. Similarly, Abdou et al. [2021] show that a model trained on text can recover key geometry of color space, even without grounding; with a few examples of grounding, LLMs are able to align their structure with the real grounded one, suggesting that they already possess the right relations Patel and Pavlick [2021]. Importantly, as the performance of LLMs has improved in recent years the extent to which their relational geometry reflects human data has also consistently increased [Peters et al., 2018, Devlin et al., 2018, Brown et al., 2020]. Larger models also better reflect the human tendency for semantic or mental models to influence formal reasoning behaviour [Wason and Johnson-Laird, 1972], on challenging logic problems that are not observed in their training data [Dasgupta et al., 2022]. Moreover, this increasing correspondence between LLMs and human data is not observed only behaviourally. Recent fMRI studies show that the semantic models that best account for the representational geometry and processing activity of human brains are precisely the neural network LLMs which are trained on the largest amount of data [Schrimpf et al., 2021, Goldstein et al., 2022, Kumar et al., 2022].

Many of the tasks that LLMs succeed on are ones that require maintaining the right relationships between concepts. Impressively, the largest models can now devise coherent narratives [Brown et al., 2020], extend stories [Xu et al., 2020, Li et al., 2021], answer factual questions [Jiang et al., 2021] solve Winograd Schema [Kocijan et al., 2020] and resolve complex quantitative reasoning problems [Lewkowycz et al., 2022]. Increasingly, such models are even aware of the likelihood that they can answer a given question correctly; i.e. they have an explicit sense of the extent of their own knowledge [Kadavath et al., 2022]. Each of these capacities requires, in some way or another, sensitivity to conceptual roles because the required words and concepts must be used jointly together in a coherent way that mimics how humans would.

Despite these empirical successes, there are many places where these models can still be improved (for detailed analyses, see, Lake and Murphy [2021], McClelland et al. [2020], Pavlick [2022]). Lake and Murphy [2021] emphasize the need for reference, inference, better and more robust compositionality, more structure and more consistent abstract reasoning. Models trained on multimodal datasets show better match to human judgements than those trained on text alone [Hill et al., 2016, De Deyne et al., 2021]; at the same time, even multimodal LLMs are missing many aspects of a complete theory of semantics, including the ability to simulate situations in which their physical or linguistic behaviour affects their environment [McClelland et al., 2020] as well as knowledge of the goals and desires that drive how people use words [Bisk et al., 2020, Lake and Murphy, 2021].

Our claim, then, is not that LLMs perfectly capture human concepts or perfectly reflect human meaning. Unlike the more radical perspectives entertained by Wittgenstein or Quine, we also do not

consider that reference should play *no* role in a principled treatment of meaning. Instead, we find it productive to consider reference as just one (optional) aspect of a word's full conceptual role. It is relevant for some concepts [see Putnam, 1974] and not others—just like color, valence, or teleology is relevant for some concepts and not others. Experience of both agency and a perceptual environment similar to our own may lead to the richest, most human-like understanding of language in machines [Bisk et al., 2020, McClelland et al., 2020].

As these improvements are made, the models will come into closer alignment with humans, and each such improvement will enrich the model's sense of meaning. This process of progressive enrichment is also found in human concepts. When people discovered that $H_2O$ was the chemical composition of water, they grew their conceptual network and even revised their reference for the term. There was no hard transition from a meaningless concept of "water" to a meaningful one. Some meaning was there all along because "water" had a conceptual role even before its chemical composition was known. What changed was the richness and interconnection of this concept—the way in which it was related to other concepts like "hydrogen" and "oxygen." In much the same way, we see no reason to assume that the world of a system that receives input from a single sensory modality is meaningless, even if the addition of further sensors provides clear enrichment. When thinking about improving LLMs it we should therefore consider ways to enrich the internal conceptual roles of these systems, including to better reflect the structure, inference and algorithmic sophistication of humans [Tenenbaum et al., 2011, Lake and Murphy, 2021, Rule et al., 2020].

**Discovering conceptual role**    Conceptual role theory also provides a compelling way to understand learning, including the way in which neural networks may come to represent symbolic processes. A symbol like $AND$ only means logical conjunction if it interacts (composes) with others symbols like $TRUE$ and $FALSE$ in the appropriate way—i.e. when it has the right conceptual role in the broader system of symbols. The technique of *church-encoding* in mathematical logic [see Pierce, 2002] provides a way to understand how such roles may be learned within neural networks or dynamical systems Piantadosi [2021]. In church-encoding, a representation is constructed in one system (e.g. lambda calculus or a neural network) in order to mimic the behavior of another system (e.g. boolean logic) in the sense that the representations in the first system interact with each other in a way that yields the desired conceptual roles of the second. Piantadosi [2021] shows how a church-encoding learner could acquire structures like logic, lists, trees, hierarchies, numbers, quantifiers, and recursion without possessing them to start, and how this metaphor may provide an "assembly language" that translates from symbolic computational or cognitive theories into underlying implementations. In this view, neural networks would train their parameters so that their internal, intrinsic dynamics church-encode the conceptual roles of a targeted domain.

The key question for LLMs is whether training to predict text could actually support discovery of conceptual roles. To us the question is empirical, and we believe has been answered in a promising, partial affirmative by studies showing success on tasks that require knowledge of relationships between concepts. Text provides such clues to conceptual role because human conceptual roles generated the text. Analogously, it is possible to build a theory of gravity from measurements of the moon's movement because gravity *generated* these movements; the goal of essentially all inductive learning techniques is to invert from observations to likely generating processes or parameters. Moreover, the entailment relationships between sentences are often intrinsically related to analogous patterns in thought [e.g. Fodor and Pylyshyn, 1988], meaning that a model which captures how sentences relate to each other might indeed capture how thoughts relate to each other. One helpful analogy is that of *embedding theorems* in dynamical systems [e.g. Packard et al., 1980, Takens, 1981] which allow some properties of systems to be recovered from seemingly impoverished representations of their state. In the paper "Geometry from a Time Series", Packard et al. [1980] show, remarkably, that one can sometimes reconstruct the geometry of a multi-dimensional dynamical system from a *one*-dimensional projection of its state. Thus, information about high-dimensional state (sometimes essentially all of it) can be decoded from the trajectory of low-dimensional projection. People use concepts in thinking and reasoning based on their meaning, and text is a low-dimensional projection of some of these patterns of use, so it is plausible that some properties of the real meaning could be inferred from text. At the very least, embedding theorems illustrate that there may not be a simple way to intuit what a learner can or cannot deduce about the underlying mental states from text alone.

The protein folding neural network AlphaFold [Jumper et al., 2021] provides further evidence that transformer-based networks can infer and generalise complex latent multi-entity structures. AlphaFold

is trained to predict the configuration of single proteins only, but acquires actionable knowledge of concepts not explicitly present in its training data. Despite never seeing a zinc ion, AlphaFold often perfectly infers its location and places all the protein side chains correctly right around it. Some proteins only fold with multiple copies of themselves (homomers). Again, AlphaFold has never seen more than one copy of a protein, but often infers both the number of required copies and their relative placement correctly. This suggests that the process of predicting the structure of single proteins enabled the AlphaFold network to infer non-trivial facts about chemistry and biology.

**Communicative intentions**   Separable from meaning and reference, many have also rejected the idea that LLMs produce language with *intention* [Bender et al., 2021, Bender and Koller, 2020]. Because LLMs are trained only on sequence prediction, they are argued to be, "stochastic parrots" Bender et al. [2021] or just sophisticated "auto-complete" algorithms[1] that cannot access the intentions of those who produced their training data, and themselves produce language without intending anything in particular. But in our view, a key difference between autocompleting parrots and LLMs is that the latter have rich, causal, and structured internal states.

One view of intent is semantic, corresponding to whether the language they produce arises from an internal representation of (intended) meaning. The conversion of internal states into language and back *is* the essential function of LLMs, embedded in their architecture and training. We have argued that LLM's internal state has some notions of conceptual role, so LLM's utterances have the semantic intent corresponding to these roles.

Another view of intent is pragmatic, asking what might be achieved by producing a sentence. This corresponds to asking whether they engage in any goal-directed *planning* [Russell, 2010] when producing language. We consider it probable that multi-layer LLMs do, in an emergent, implicit sense, execute a form of planning as part of the process of repeated (self-attention-based) analyses of current and past inputs. These computations likely involve representation of the current situation and at least implicit evaluation of consequences of utterances. Recent work has argued that LLMs posses model-like belief structures Hase et al. [2021] and update representations of dynamic semantics, objects and situations, throughout a discourse Li et al. [2021]. These emergent semantics causally determine LLM output. Of course, differences between humans and LLMs in their training experience and objectives mean that the planning process in LLMs is less explicit and sophisticated than in humans (making errors more likely, for instance, in cases of hypothetical or counterfactual reasoning [Ortega et al., 2021]).

**Conclusion**   Bender & Koller argue that text-based LLMs will never have meaning because these models lack reference. However, they do not demonstrate that reference is the key to meaning— instead they assume it. As we have argued, this assumption is hard to reconcile with theories of cognition and the phenomena that motivate them. People are happy to think about concepts without referents and otherwise often don't know many details of reference. Meaning instead seems to come from the way concepts relate to *each other*. It is these interrelations that LLMs know something about since their internal geometries and trajectories approximate those of humans. Like people who don't know that water is $H_2O$ and so could not pick it out based on chemical composition, Bender & Koller's octopus lacks some aspects of conceptual role like physical appearance. But, both the octopus and people know other parts of conceptual role that are sophisticated in their own right. If theories about conceptual role are the correct account, then LLMs likely already share the foundation of how our own concepts get their meaning.

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
