# OpenReview forum: "Meaning without reference in large language models"
_NeurIPS.cc/2022/Workshop/nCSI — nCSI WS @ NeurIPS 2022 Oral_

### Official Review · Reviewer_Gz7Q · 2022-10-15

**Rating:** 2
**Confidence:** 2

**Review:**

**Summary of the Paper**

Recent works Indicate that **large language models** (LLM) are incapable of inferring meaning as they do not have any reference object to relate the text to, citing examples such as the octopus test. The paper refutes this stating this to be a singular perspective of meaning and introduces another perspective -Conceptual Role Theory -  where the meaning of a word is rather associated with the context of the words surrounding it thus rendering large language models capable of inferring meaning.

**Main Review**

**Pros**

- The paper is very well written and easy to follow. The authors motivate the problem by providing sufficient details of the current literature and their singular viewpoint. Furthermore, the authors provide sufficient details of the theoretical background such as Conceptual Learning Theory among others -needed to understand the thesis - which makes the paper self sufficient.
-  The paper does not claim that LLM's can learn human level concepts but rather claim that reference should not play a role in determining meaning which is appropriate.

**Cons**

- The section "Communicative Intentions" of the paper states that the internal states of LLM's capture rich, causal and structural information. While rich and structural have been proven to exist, causal is still an open avenue. An example I would like to cite is the paper  'Attention is not explanation' where they demonstrate that there exists a set of counterfactual attention weights which produce the same output as the learned weights do.
- While the paper provides examples of discovering conceptual roles in the context of learning in several large models such as Alphafold, I would have liked to see the example of an LLM which would have affirmed my understanding of the section w.r.t to LLM's

**Typos**

Line 231 posses --> possess

**Summary of the Review**

Overall, the paper is an interesting read and easy to follow. It is also well motivated and sufficient details are provided to make the paper self sufficient. Certain statements such as LLM's capture causal structure can be further revised based on other papers referenced in this review. Some minor typos could be removed

---

### Official Review · Reviewer_ut7i · 2022-10-15
**The authors discuss an intersting approach of concept understanding and meaning for LLMs.**

**Rating:** 3
**Confidence:** 2

**Review:**

## Summary
The paper is concerned with the fundamental problem of whether concepts of the real world can be understood (exclusively) from text descriptions.
In particular the authors argue against the position presented in "Climbing towards NLU: On meaning, form, and understanding in the age of data" (Bender and Koller [2020]), in which it is argued that learning systems a priory cannot learn meaning of real-world concepts from text corpora.
Bender and Koller argue, e.g., with the example of the "octopus test", that learning the use of the word "coconut" and its relations from text correlation does not help an intelligent agent in identifying a coconut in the real world as there is no mapping from the textual concept to the real world entity or its related real world concepts. The authors argue that by such logic we would not be able to understand abstract or impossible concepts (e.g., "contract" / "perpetual motion machines") as there is no real world reference, while humans clearly do think of, and understand the meaning of such concepts.
The authors discuss the conceptual role theory in the context of LLMs (and other learning systems) and argue that the meaning of concepts is not only defined by observing correlation of the same concept across different modes (text+visual), but rather by the embedding of a concept (e.g., via church-encodings) into a network of related concepts within their domain.


## Relevance
The paper is relevant to the workshop as it discusses the fundamental question of whether machines can learn meaning about real world concepts from textual descriptions without collecting additional experience e.g., in the form of images. The discussion relates closely to the task of (causal) structure discovery and discusses the conditions under which machine learning systems may be able to learn meaningful concepts by relating those concepts to each other and across domains.


## Related work
While I'm not familiar with the field of conceptual role theory the paper seems to thoroughly consider and discuss relevant literature throughout the paper.


## Strengths
* The authors extended the definition by which meaning has to be observed from correlations over different modes of presentation. Their novel extended definition allows the identification of meaning from the embedding structure of their related concepts, which overcomes the criticism of non-identifiability from the initial, purely correlational, definition.
* The authors convincingly argue that projections into lower dimensional representations, e.g. onto time series, allow an agent to identify concepts by the structure of their embeddings across modes of presentation.


## Weaknesses
* While the authors provide a thorough discussion the paper might benefit from formalizing the discussed topics. This would help to analyze the specific preconditions needed to allow or prevent concept identification across different modes. In this context the authors might also consider connections to ideas in [1] and [2].
* While the authors provide a novel way of defining and identifying meaning in the context of LLMs it would be helpfull if the authors could provide an exemplary proof of identifying in the octopus example.
* The conclusion contains an unjustified generalization: "people are happy to think [...] often don't know many details" - which I consider to be vague and unsubstantiated. "happy to think" implies naivety, while "often don't know many details" is a vague and unproven statement. I would recommend to tone down this particular statement. For example, the authors might consider to restrict their criticism to the specific set of reviewed papers; and might want to list the specific missing "details", which in the first place seem to be a too narrow definition of reference in comparison to their papers' definition.


## Conclusion
The authors convincingly argue that concept identification across different modes of observation is possible by taking advantage of shared representations in meta-level domains, such time-series or logic, to recover conceptual role. They further argue that using structure recovery methods allows the reconstruction of concepts from their lower-dimensional projections and match them across different modes of observation. While these results may hold in general between arbitrary domains, the authors specifically discuss the implications for LLMs of learning the meaning of real-world concepts from text representation. While the paper could benefit from a more formal definition of the proposed concepts I consider the presented work a valuable addition to the workshop as well as to the general debate about whether MLs/LLMs are capable of developing understanding of the real-world from text.


[1] Gresele, Luigi, et al. "Causal inference through the structural causal marginal problem." *International Conference on Machine Learning*. PMLR, 2022.

[2] Pfister, Niklas, and Jonas Peters. "Identifiability of Sparse Causal Effects using Instrumental Variables." *arXiv preprint arXiv:2203.09380* (2022).

---

### Meta-Review · Area_Chair_DMVD · 2022-10-17

**Recommendation:** 3
**Confidence:** 3

**Metareview:**

The paper revisits the question of whether LMs can learn meaning. Actually it argues that they are  likely to capture important aspects of meaning. To this aims it makes use of the conceptual role theory from cognitive science, arguing that meaning can only be determined by examination of how a model’s internal states relate to each other. Injecting the conceptual role theory into the discussion about meaning and LMs is important, though some empirical results would have been great. Here, it might be interesting to link to relative representations found in deep networks (https://arxiv.org/abs/2209.15430). Overall, both reviewers agree that the paper makes an interesting and important contributions and that it should be accepted, though the connections to causality could be improved.

---

### Decision · Program_Chairs · 2022-10-20

Accept (Oral)